# Pathogen Detection and Resistome Analysis in Healthy Shelter Dogs Using Whole Metagenome Sequencing

**DOI:** 10.3390/pathogens14010033

**Published:** 2025-01-05

**Authors:** Smriti Shringi, Devendra H. Shah, Kimberly Carney, Ashutosh Verma

**Affiliations:** 1School of Veterinary Medicine, Texas Tech University, Amarillo, TX 79106, USA; devendra.shah@ttu.edu; 2College of Veterinary Medicine, Lincoln Memorial University, Orange Park, FL 32073, USA; kimberly.carney@lmunet.edu; 3Center for Infectious, Zoonotic and Vector-Borne Diseases, Lincoln Memorial University, Harrogate, TN 37752, USA; 4Richard A. Gillespie College of Veterinary Medicine, Lincoln Memorial University, Harrogate, TN 37752, USA

**Keywords:** metagenomics, shelter dogs, antimicrobial resistance, pathogen

## Abstract

According to the Humane Society, 25 to 40 percent of pet dogs in the United States are adopted from animal shelters. Shelter dogs can harbor bacterial, viral, fungal, and protozoal pathogens, posing risks to canine and human health. These bacterial pathogens may also carry antibiotic resistance genes (ARGs), serving as a reservoir for antimicrobial resistance (AMR) transmission. This study aimed to utilize whole metagenome sequencing (WMS) to screen for microbial pathogens and assess the resistome in healthy shelter dogs. Fecal samples from 58 healthy shelter dogs across 10 shelters in Kentucky, Tennessee, and Virginia were analyzed using WMS. Genomic DNA was extracted, and bioinformatics analyses were performed to identify pathogens and ARGs. The WMS detected 53 potentially zoonotic or known pathogens including thirty-eight bacterial species, two protozoa, five yeast species, one nematode, four molds, and three viruses. A total of 4560 ARGs signatures representing 182 unique genes across 14 antibiotic classes were detected. Tetracycline resistance genes were most abundant (49%), while β-lactam resistance genes showed the highest diversity with 75 unique ARGs. ARGs were predominantly detected in commensal bacteria; however, nearly half (18/38, 47.4%) of known bacterial pathogens detected in this study carried ARGs for resistance to one or more antibiotic classes. This study provides evidence that healthy shelter dogs carry a diverse range of zoonotic and antibiotic-resistant pathogens, posing a transmission risk through fecal shedding. These findings highlight the value of WMS for pathogen detection and AMR surveillance, informing therapeutic and prophylactic strategies to mitigate the transmission of pathogens among shelter dog populations and the risk associated with zoonoses.

## 1. Introduction

In the United States, 65 million households owned a pet dog in 2023 [1]. Interestingly, the US Humane Society report shows that over a quarter of these dogs were adopted from animal shelters [2]. The animal shelters typically admit dogs irrespective of age, breed, sex, and source or health status, with over 40% of their intake comprising stray or abandoned animals that often present with poor health [3]. The operational policies and procedures of animal shelters vary widely, with some shelters implementing strict quarantine and pre-transfer medical requirements, whereas most others have no such requirements. Additionally, due to the high traffic and the intermixing of animals from different sources and medical backgrounds, shelter dogs are frequently exposed to various infectious agents and often experience a variety of stressors that further increase their susceptibility to infectious agents.

Most published epidemiologic studies focused on targeted, pathogen-specific surveillance in the United States show that shelter dogs can harbor pathogens. For instance, a few studies have reported that shelter dogs can be carriers of zoonotic pathogens such as *Campylobacter* spp. (13% to 75% prevalence) or *Salmonella* spp. (1.9% to 8.3% prevalence) [4,5,6,7]. We previously reported the fecal prevalence of *Campylobacter* (13.6%, 8/59) and urinary shedding of *Leptospira* (13.1%, 26/198) among the shelter dog population in the tri-state area of western Virginia, eastern Tennessee, and southeastern Kentucky [8,9]. From nasal and perianal swab cultures collected at a dog shelter in Colorado, methicillin-resistant *Staphylococcus pseudintermedius* (MRSP) was detected in one out of two-hundred dogs (0.5%) and methicillin-sensitive *S. pseudintermedius* in six out of two-hundred dogs (3%) [10]. Gastrointestinal parasites such as *Ancylostoma caninum* (20.7% to 26.4%) and *Trichuris vulpis* (12% to 14.1%) were detected from fecal samples collected from 529 shelter dogs in Texas [11]. In Oklahoma, *Trypanosoma cruzi* infection in shelter dogs was detected by PCR [three of one-hundred and eighty-nine (1.6%)] and via sero-surveillance [twenty-six of one-hundred and ninety-seven (13.2%)] [12]. Moreover, several vector-borne infectious agents were identified in shelter dogs, including *Anaplasma phagocytophilum*, *Anaplasma platys*, *Babesia* sp. *(Coco)*, *Borrelia burdgoferi*, *Dirofilaria immitis*, *Ehrlichia canis*, *Ehrlichia chaffeensis*, *Ehrlichia ewingii*, *Rickettsia rickettsii*, and *Hepatozoon americanum* [13]. Similarly, up to 33% of the dogs from multiple shelters at the western US–Mexico border were reported to be infected with tick-borne pathogens, including *Rickettsia rickettsii*, *Echrlichia canis*, and *Anaplasma platys* [14]. Recently, a large-scale study conducted from May 2014 to June 2020 reported that 1393 (23.5%) of the 5920 sheltered dogs housed in 98 different animal welfare organizations in the United States tested positive for canine distemper virus (CDV) [15]. These studies collectively show that healthy shelter dogs harbor a variety of zoonotic and canine pathogens, highlighting their potential as reservoirs for microbial transmission to humans (shelter workers and adopters), other animals, and among shelter dogs.

Most pathogen surveillance studies in shelter dogs have focused on detecting a limited number of pathogens using traditional culture, serological, or molecular techniques. However, these targeted pathogen detection approaches fail to capture the full range of microbial pathogens that shelter dogs may harbor which are significant to both animals including canine and public health. This limitation increases the risk of pathogen transmission from carrier dogs to other animals or humans. Given the high turnover and movement of shelter dogs between shelters and households, comprehensive pathogen screening is crucial for understanding the full pathogen load of individual dogs. Such screening can inform the design of targeted therapeutic and preventive strategies, reducing the risk of outbreaks in shelters. Additionally, it can help guide decisions regarding adoption suitability/pre-adoption interventions, particularly for families with high-risk individuals, such as young children (<5 years), older adults (>60 years), or those with underlying health conditions. In this study, we employed whole metagenome sequencing (WMS) as a comprehensive tool to detect a wide spectrum of canine and zoonotic pathogens, as well as the antimicrobial resistome, in fecal samples collected from shelter dogs.

## 2. Materials and Methods

### 2.1. Sampling

A total of 58 healthy dogs (35 male and 23 female) from ten shelters (coded as AB, BC, BR, CC, JO, KR, KW, LC, SL, and UC) in the Cumberland Gap Region of Kentucky, Tennessee, and Virginia were selected for this study. The age of the dogs ranged from <1 year (*n* = 17), 1 to 2 years (*n* = 23), and >2 years (*n* = 18). In the summer of 2019, freshly voided fecal samples (5–50 g) were collected from each dog in a sterile bag, immediately placed on ice, and shipped to the laboratory. The fecal samples were stored at −80 °C until further processing for DNA extraction. The genomic DNA was extracted from each fecal sample using the QIAamp DNA Stool Mini Kit (Qiagen, Germantown, MD, USA). Informed consent was obtained from the animal shelter directors for the collection of the freshly voided fecal samples. The collection of these samples was exempted by the Institutional Animal Care and Use Committee (IACUC) at the Lincoln Memorial University.

### 2.2. Whole Metagenome Sequencing (WMS)

The total genomic DNA was extracted from the individual fecal samples using QIAamp DNA Stool Mini Kit (Qiagen, Germantown, MD, USA) following the manufacturer’s instructions. The individual DNA samples were sequenced and analyzed using the ZymoBIOMICS shotgun sequencing pipeline (Zymo Research, Irvine, CA, USA) and ZymoBIOMICS bioinformatics analysis pipeline. Briefly, genomic DNA samples were profiled with shotgun metagenomic sequencing. Sequencing libraries were prepared with the Nextera^®^ DNA Flex Library Prep Kit (Illumina, San Diego, CA, USA) with up to 100 ng DNA input following the manufacturer’s protocol using internal dual-index 8 bp barcodes with Nextera^®^ adapters (Illumina, San Diego, CA, USA). The ZymoBIOMICS^®^ Microbial Community DNA Standard (Zymo Research, Irvine, CA, USA) was used as a positive control for each targeted library preparation. Negative controls (i.e., blank library preparation control) were included to assess the level of bioburden carried by the wet lab process. All libraries were quantified with TapeStation^®^ (Agilent Technologies, Santa Clara, CA, USA) and then pooled in equal abundance. The final pool was quantified using qPCR. The final library was sequenced (paired-end, 2 × 150 bp) on the Illumina NovaSeq^®^ (Illumina, San Diego, CA, USA).

### 2.3. Data Analysis

The raw sequence reads from each sample were analyzed using the ZymoBIOMICS bioinformatics analysis pipeline. Briefly, raw sequence reads were trimmed to remove low-quality fractions and adapters with Trimmomatic-0.33 [16]: quality trimming was done by a sliding window with a 6 bp window size and a quality cutoff of 20 and reads with size lower than 70 bp were removed. The microbial composition was profiled with Centrifuge [17] using bacterial, viral, fungal, mouse, and human genome datasets. Strain-level abundance information was extracted from the Centrifuge outputs for downstream analysis including (1) alpha- and beta-diversity; (2) microbial composition with QIIME [18]; and (3) taxa abundance heatmaps with hierarchical clustering (based on Bray–Curtis dissimilarity). Antimicrobial resistance genes (ARGs) were identified with the DIAMOND sequence aligner [19]. The data from ARG and microbial composition profiling were extracted to an excel file. All the ARGs were then manually assigned to a specific class of antibiotics. The microbial community data were screened to identify individual pathogens and categorized into four major pathogen groups: (i) known canine pathogens, (ii) known/potentially zoonotic pathogens, (iii) opportunistic pathogens of canines or humans, and (iv) pathogens of other animals. The outcomes of the shotgun metagenomics were described using descriptive statistics to summarize the results of microbiome diversity, the prevalence of pathogens, and the resistomes (ARGs) in shelter dogs. The differences in microbial communities between sexes, age groups, and shelters were compared using the Shannon diversity index by one-way ANOVA with Tukey’s post hoc using NCSS 2023 version 23.0.1 (NCSS, Kaysville, UT, USA).

## 3. Results and Discussion

### 3.1. Fecal Microbiome Diversity and Composition in Shelter Dogs

From the WMS, a total of approximately 565 million raw reads were obtained, with an average of 9.75 million reads per sample (Appendix A). After the quality trimming and removal of host reads and unclassified reads, a total of 135 million reads were identified as microbial reads, with an average of 2.32 million reads per dog (range = 0.7 to 4.87 million reads from individual dogs).

Bacterial taxonomic units (BTUs) predominated in all samples (Appendix A), constituting 99.5% of the fecal microbiome, contributing to a greater phylogenetic diversity compared to eukaryotes (0.06%) and viruses (0.46%). These results corroborate with previous gut microbiome studies in dogs [20,21]. The sequenced reads were taxonomically assigned to 12 bacterial phyla [*Acidobacteria* (0.003% BTUs, *n* = one species), *Actinobacteria* (1.98% BTUs, *n* = 63 species), *Bacteroidetes* (70.63% BTUs, *n* = 110 species), *Chlamydiae* (0.06% BTUs, *n* = two species), *Chloroflexi* (0.001% BTUs, *n* = one species), *Deferribacteres* (0.01% BTUs, *n* = one species), *Deinococcus-Thermus* (0.08% BTUs, *n* = three species), *Firmicutes* (10.73% BTUs, *n* = 235 species), *Fusobacteria* (1.44% BTUs, *n* = seven species), *Proteobacteria* (14.93% BTUs, *n* = 86 species), *Spirochaetes* (0.14% BTUs, *n* = six species), and *Verrucomicrobia* (0.0004% BTUs, *n* = one species)]; six Eukaryote phyla [*Apicomplexa* (n = one species), *Ascomycota* (*n* = seven species), *Basidiomycota* (*n* = three species), *Discosea* (n = one species), *Mucoromycota* (*n* = one species), and Nematoda (*n* = one species)]; three pathogenic viral genera [canine minute virus, canine parvovirus, and an enterovirus]; and 10 bacteriophages [*Escherichia*, *Shigella*, *Salmonella*, *Bacteroides*, *Enterococcus*, *Faecalibacterium*, *Lactobacillus*, *Parabacteroides*, *Stenotrophomonas*, and *Streptococcus* phages and one unknown virus] (Figure 1). In concordance with previous studies [20,21], the predominant bacterial phyla in individual dogs included *Bacteroidetes* (68.7% OTUs, 95% confidence interval (73.3%, 64%)), *Proteobacteria* (15.7% OTUs, 95% confidence interval (19.35%, 12%)), and *Firmicutes* (11.6% OTUs, 95% confidence interval (13.7%, 9.6%)), these being the most abundant, followed by *Actinobacteria* (2.4% OTUs, 95% confidence interval (3.4%, 1.36%)) and *Fusobacteria* (1.3% OTUs, 95% confidence interval (2.3%, 0.29%)).

The Shannon diversity index (SDI) for the fecal microbiome of individual shelter dogs ranged from 0.73 (sample SL6) to 3.59 (sample JO3), with an average SDI of 2.48 (Appendix A, Figure 1). At the shelter level, SDI values ranged from 1.89 (shelter SL) to 3.02 (shelter KR). No statistically significant differences in SDI were observed between male and female dogs or between the three age groups of dogs.

### 3.2. Fecal Pathogen Prevalence in the Shelter Dog Population

WMS identified a total of 53 potential or known pathogen species (Figure 2). These included bacteria (*n* = 38 species, distributed across 18 genera), protozoa (*n* = two, *Toxoplasma gondii* and *Balamuthia mandrillaris*), yeast (*n* = five, *Candida parapsilosis*, *Malassezia globose*, *Malassezia restricta*, *Malassezia sympodialis*, and *Malassezia* spp.), nematode (*n* = one, *Trichuris trichiura*), mold (*n* = four, *Alternaria alternata*, *Fusarium fracticaudum*, *Fusarium proliferatum*, and *Saksenaea oblongispora*), and viruses (*n* = three, canine parvovirus 2, canine parvovirus 1, and an enterovirus). These pathogens were broadly classified based on their known or potential pathogenicity in dogs, humans, or other animal species, resulting in four categories: known canine pathogen, known or potential zoonotic pathogen, opportunistic pathogen for canines or humans, and other animal pathogen.

#### 3.2.1. Known Canine Pathogens

Three known canine pathogens, *Carnivore protoparvovirus 1*, *Carnivore bocaparvovirus 1*, and *Bordetella bronchiseptica*, were identified by fecal metagenome sequencing in two shelters. Shelter KR harbored all three pathogens, while shelter KW carried *Carnivore bocaparvovirus 1. Carnivore protoparvovirus 1* (also called canine parvovirus 2 or CPV-2) was detected in fecal samples from a total of three dogs (5.2%), including two (28.6%) from shelter KW (KW1 and KW3) and one from shelter KR (KR5). In shelter KR, a fecal sample from one dog (KR4) was positive for *Carnivore bocaparvovirus 1* (also called canine minute virus or CPV-1) and another dog (KR7) was positive for *Bordetella bronchiseptica*. All the dogs except KR7 were less than two years of age, with KW1 and KW3 less than a year old. CPV-2 is known to cause severe gastroenteritis, especially in young puppies of less than 6 months of age, whereas CPV-1 is known to cause a range of pathologies, including enteritis, pneumonitis, and neonatal death, in dogs [22]. Parvoviral gastroenteritis is a highly contagious viral disease that is known to pose a significant health challenge in animal shelters, often leading to mortality and significant economic burden due to the high cost of treatment and euthanasia [23]. While the fecal shedding of CPV in a few healthy dogs in a shelter can pose a significant risk of transmission to other dogs, it is currently unknown if the CPV detected via metagenomics is a virulent strain or a vaccine strain. Similarly, *Bordetella bronchiseptica*, a causative agent of contagious kennel cough, was detected in one dog (KR7). Given that the usual habitat of *Bordetella bronchiseptica* is the upper respiratory tract of dogs, its detection in fecal samples is unexpected. However, this may occur through two potential mechanisms: dogs that are coughing may swallow respiratory secretions containing the bacteria, leading to its passage through the gastrointestinal tract. Additionally, dogs frequently engage in behaviors such as sniffing or licking the perianal area of other dogs, which could introduce the bacteria into the fecal material [24].

#### 3.2.2. Known or Potentially Zoonotic Pathogens

A total of 12 species (10 bacteria, one nematode, and one protozoa) of known or potentially zoonotic pathogens were identified in this study (Figure 2). These pathogens included *Campylobacter upsaliensis*, *Campylobacter jejuni*, *Salmonella enterica*, *Helicobacter canis*, *Clostridioides difficile*, *Clostridium perfringens*, *Clostridium botulinum*, *Brachyspira pilosicoli*, *Mycobacterium tuberculosis*, *Balamuthia mandrillaris*, *Toxoplasma gondii*, and *Trichuris trichiura.* Given that the significance of each of these as animal infectious agents varies widely depending on several factors, the pathogenicity and zoonotic potential of these bacteria in animals and humans is discussed below.

*Campylobacter upsaliensis* was detected in fecal samples from 75.9% (44/58) of dogs, representing all ten shelters, whereas *Campylobacter jejuni* was detected in 20.7% (12/58) of dogs, representing six shelters. Published studies show a 50 to 90% prevalence of *Campylobacter* fecal shedding [4,5,6], with both *C. jejuni* and *C. upsaliensis* being frequently detected in the shelter dog population. *C. jejuni* is frequently implicated as a cause of canine campylobacteriosis [25], whereas *C. upsaliensis* is known to cause mild to moderate enteric disease in dogs [25,26]. Both *C. jejuni* and *C. upsaliensis* are also the most significant zoonotic pathogens in people, especially in children and the elderly [27,28], with an annual estimated incidence of 20 cases of *Campylobacter*-associated illness for every 100,000 people in the United States [29,30]. Evidence suggests that zoonotic transmission of *C*. *jejuni* occurs through direct contact with the feces of infected dogs [31,32,33]. In some cases, *Campylobacter* infection can be invasive in people, leading to severe complications including Guillain–Barré syndrome [34].

*Salmonella enterica*, an enteric zoonotic pathogen known to infect a wide variety of animals and humans, was detected in one (1.7%) of the dogs from a single (10%) shelter. One study reported fecal shedding of *Salmonella* in 4.9% (26/226) of shelter dogs in Texas [35]. A multi-laboratory survey conducted in the United States reported 2.5% (60/2422) of *Salmonella* prevalence in diarrheic and non-diarrheic dogs [36], suggesting that *Salmonella* shedding in healthy dogs is not uncommon. Although dogs can acquire *Salmonella* infection from a variety of sources, raw foods are considered major risk factors for the acquisition of infection and have caused major outbreaks involving both dogs and humans [37]. The fecal shedding of *Salmonella* in dogs poses a zoonotic risk of transmission to humans [38].

Non-*Helicobacter pylori* helicobacters (NHPH) such as *Helicobacter canis* are known to naturally colonize the gastrointestinal tract of dogs. In this study, *Helicobacter canis* was detected in 65.5% (38/58) of dogs from all ten shelters (Figure 2). Previously, *H. canis* prevalence in canine feces has been reported to range from 5.8% (23/390) in Chile and 21% (20/95) in Sweden to 100% (14/14) in Taiwan [39,40,41]. *H. canis* is occasionally reported from enterohepatic disease in dogs [42]. Moreover, close contact with dogs and cats was reported as the primary risk factor among immunocompromised human patients living with *H. canis*-infected dogs, suggesting the potential for zoonotic transmission of *H canis* from dogs to humans [43,44]. However, the pathogenicity of *H. canis* in dogs and its zoonotic potential is likely underrecognized, most likely due to its fastidious nature that poses challenges in culturing, likely leading to false-negative cultures.

*Clostridioides difficile* (previously known as *Clostridium difficile)* was detected in a total of ten (17.2%) dogs, representing nine shelters, *Clostridium perfringens* was detected in eleven (19%) dogs from eight shelters, and *Clostridium botulinum* was detected in six (10.3%) dogs, representing four shelters (Figure 2). While the virulence profiling of pathogens is beyond the scope of this study, none of the *C. difficile* positive fecal samples were positive for the presence of the binary toxin gene (*cdtB*) (data not shared). The prevalence of toxigenic strains of *C. difficile* in dogs has been reported to range from 17% in healthy adult dogs [45] to 90% in puppies [46,47] and from 19 to 33% in outpatients and hospitalized dogs that are treated with antibiotics [48,49,50,51]. Dogs that visit human healthcare facilities, interact with hospitalized children, or live with an owner treated with antibiotics show an increased risk of *C. difficile* fecal shedding [52]. Thus, the zoonotic transmission of *C. difficile* between dogs and people is likely; however, a significant knowledge gap remains in our understanding of the role of *C. difficile* in enteric disease in dogs, primarily due to the lack of a concrete correlation between *C. difficile* infection and enteric disease [53]. *C. perfringens* is also frequently detected in the feces of healthy dogs and dogs with acute hemorrhagic diarrhea syndrome. *Cpa* gene (encoding phospholipase C toxin)-positive *C. perfringens* that also carry other toxin genes are more likely to be associated with disease in dogs [9,54,55]. Although screening for all potential toxin-encoding genes for *C. perfringens*-positive fecal samples from shelter dogs was beyond the scope of this study, four (AB4, JO2, KR5, and KW2) out of eleven *C. perfringens*-positive fecal samples sequenced in this study were also positive for the *Cpa* gene. Nine out of eleven *C. perfringens*-positive fecal samples sequenced in this study were previously reported as *Cpa* gene-positive by PCR [9]. While the role of toxin-positive *C. perfringens* as a potential zoonotic pathogen remains poorly recognized, *Clostridium botulinum*, a known human pathogen, was detected in fecal samples from a total of six (10.3%) dogs from three shelters including LC (*n* = three), UC (*n* = one), SL (*n* = one), and KW (*n* = one). Intestinal colonization of *C. botulism* has been reported in naturally infected and experimentally inoculated dogs without clinical signs of the disease [56]. Moreover, *C. botulism* infection with clinical disease has been linked to eating poorly stored dry food [57] and rotten bovine [58], pigeon [59], or duck [60] carcasses. The likely sources of *C. botulinum* infection in shelter dogs tested in this study remain unknown; however, the fecal prevalence of *C. botulinum* in shelter dogs poses a risk of the persistent transmission of infection among dogs. Zoonotic transmission of *C. botulinum* from dogs to people has not been conclusively demonstrated; however, the risk cannot be ruled out because *C. botulism* is also known to be transmitted to humans by a variety of transmission routes [61].

*Brachyspira pilosicoli*, a causative agent of intestinal spirochetosis in various animal species including dogs, pigs, poultry, rodents, and humans [62,63], was detected in the fecal samples of three (5.2%) dogs from two (20%) shelters. Although *B. pilosicoli* is commonly detected in the feces of dogs and has been sporadically isolated from cases of canine diarrhea [62,64], its role as a potential pathogen in dogs is yet to be conclusively determined. Comparative analyses of *B. pilosicoli* isolates from several species, including dogs and humans, suggest *B. pilosicoli* has a clear zoonotic potential [63,65,66].

The fecal samples of two (3.4%) of dogs from two (20%) shelters examined in this study were positive for *Mycobacterium tuberculosis*. While humans are major reservoir hosts of *M. tuberculosis*, dogs often serve as sentinels of human infection within households and can develop both pulmonary and disseminated forms of tuberculosis [67,68]. *M. tuberculosis* infection in dogs following exposure to infected humans (reverse zoonoses) and transmission from dogs to humans has been reported in many parts of the world, with sporadic reports from the United States [69,70,71].

Besides bacterial pathogens, the fecal samples of four (6.9%) dogs from two (20%) shelters tested positive for *Balamuthia mandrillaris*, a free-living amoeba that causes encephalitis in competent and immunocompromised humans, horses, dogs, sheep, and nonhuman primates. Amoebas are found in soil and are likely transmitted through the inhalation of airborne cysts or by the direct contamination of skin wounds. A few cases of *Balamuthia* amoebic encephalitis in dogs have been reported in the United States [72,73]. However, its true incidence in dogs and zoonotic potential from dogs to humans remains poorly understood, likely because the clinical signs often mimic other forms of encephalitis and only a few laboratories perform appropriate diagnostics [74].

*Toxoplasma gondii*, an Apicomplexan protozoan that infects almost all warm-blooded animals including livestock, birds, cats, dogs, and humans, was detected in seven (12.1%) dogs from five shelters. Interestingly, dogs do not produce infective oocysts of *T. gondii*, but rather acquire infection via the ingestion of infectious oocysts from the environment contaminated with cat feces leading to seroconversion and, in some cases, clinical disease [75]. Most infected dogs can shed infectious oocysts in their feces and mechanically transport infection to humans and other animals via contaminated body surfaces, the mouth, and the feet. Thus, dogs not only serve as vehicles for the zoonotic transmission of infection to humans but also as sentinels for environmental contamination with infective oocysts [76].

The intestinal nematode *Trichuris trichiura* (whipworm), a causative agent of human trichuriasis, was detected in eight (13.8%) dogs from five (50%) shelters. Trichuriasis is a neglected tropical disease that affects >800 million people worldwide, including in the southern United States [77]. The primary hosts of *T. trichiura* are humans and non-human primates. Canine whipworm infections are generally considered to be *T. vulpis.* However, *T. trichiura* eggs have been occasionally detected in stool samples from dogs [78]. Although zoonotic infections with other *Trichiura* spp. such as *T. suis* (from pigs) and *Trichiura vulpis* (from dogs) have been reported in humans [79], potential zoonotic transmission of *T. trichiura* from shelter dogs to humans or between dogs remains unknown. The detection of *T. trichiura* in multiple dogs from multiple shelters in this study suggests a rather high prevalence which warrants further investigations on potential zoonotic risks. Finally, the signatures of Enterovirus C (human poliovirus 1 Mahoney) were detected in six (10.3%) dogs from four (40%) shelters. Interestingly, human enteroviruses, including polioviruses, have been isolated from dog feces previously [80,81].

#### 3.2.3. Opportunistic Pathogens (Canine and Human)

Twenty-five opportunistic pathogen species, including seven genera of bacteria, three genera of mold, two genera of yeast, and an enterovirus, were identified in shelter dogs. Seven opportunistic pathogens, including *Enterococcus faecium* (46.6% of dogs and 100% of shelters), *Enterococcus hirae* (15.5% of dogs and 60% of shelters), *Escherichia coli* (51.7% of dogs and 90% of shelters), *Staphylococcus aureus* (8.6% of dogs and 40% of shelters), *Streptococcus gallolyticus* (51.7% of dogs and 80% of shelters), *Malassezia restricta* (8.6% of dogs and 50% of shelters), and an enterovirus (10.3% of dogs and 40% of shelters), were commonly detected. Other opportunistic pathogens, including *Proteus mirabilis*, *Staphylococcus epidermidis*, *Streptococcus pasteurianus*, *Fusarium fracticaudum*, *Saksenaea oblongispora*, and *Malassezia globosa*, were detected from two to three dogs from up to three shelters. Opportunistic pathogens, including *Enterobacter cloacae*, *Enterobacter ludwigii*, *Enterococcus faecalis*, *Pasteurella multocida*, *Streptococcus canis*, *Streptococcus mitis*, *Streptococcus sinensis*, *Alternaria alternata*, *Fusarium proliferatum*, *Candida parapsilosis*, *Malassezia* sp., and *Malassezia sympodialis*, were detected, representing one dog each from eight different shelters. Also noteworthy is that the opportunistic pathogens listed here are also known to be a component of the normal canine gut microflora; thus, the detection of these organisms in fecal samples can be expected and this finding does not necessarily signify as primary concern for infection. However, several of these organisms have been frequently reported as a cause of secondary opportunistic infection in both animals and humans. A range of underlying health conditions—such as cancer, wounds, urinary calculi, heart disease, and stressful situations—can predispose individuals to secondary infections caused by these microbes.

#### 3.2.4. Other Animal Pathogens

Thirteen species belonging to seven genera of bacteria were characterized as other animal pathogens. *Chlamydia abortus* was detected in all the shelters, with 81% of dogs positive, in this study. *Clostridium spiroforme* was detected in 25.9% of dogs from 60% of the shelters investigated in the study.

Collectively, these findings underscore the diverse and complex fecal pathogen landscape within the shelter dog population, encompassing a wide range of pathogens of known and potential significance to both canine and human health. This study shows that WMS can be effectively employed to detect and monitor a battery of zoonotic and other infectious pathogens from fecal samples from shelter dogs. The WMS-based detection of obligate pathogen signatures in fecal samples from healthy dogs in this study likely suggests inapparent carrier infections. The detection of opportunistic pathogen signatures from healthy dogs must be inferred with caution unless supported by follow-up investigations, including clinical findings, culture-based results, or immune responses from individual animals or shelter populations [82,83].

### 3.3. Fecal Resistome in Shelter Dog Population

The WMS of fecal samples from 58 shelter dogs yielded a total of 48,267 ARG-specific reads, resulting in the identification of 4560 ARG signatures spanning 182 unique ARGs encoding resistance to 14 different antibiotic classes, including disinfectants such as quaternary ammonium compounds (QUATs) (Appendix A). The abundance of ARG-specific reads in individual dogs ranged from 0.4% (UC3) to 4.2% (LC6) (Appendix A). ARGs encoding resistance to tetracycline, β-lactamase, macrolide, and lincosamide were detected in fecal samples from all (100%) dogs tested in this study (Figure 3). The fecal samples from the majority of dogs also tested positive for ARGs encoding resistance to pleuromutilin, lincosamide, and streptogramin A (PLSAs, 74.1%), mupirocin (75.9%), glycopeptides (87.9%), and aminoglycosides (82.8%). A few dogs tested positive for ARGs encoding resistance to sulfonamides (25.9%) and chloramphenicol (13.8%), whereas the ARGs encoding resistance to fluoroquinolones (5.2%), QUATs (5.2%), rifampicin (1.7%), and trimethoprim (1.7%) were rarely detected (Figure 3). On average, the fecal samples of shelter dogs tested in this study harbored 28.5 unique ARGs (range = 15 to 43 ARGs/dog) encoding resistance against six to ten antibiotic classes (Figure 4). Notably, the most diverse array of ARGs (75 unique ARGs) encoded resistance to the β-lactam class of antibiotics followed by the number of unique ARGs encoding resistance to tetracyclines, glycopeptides, aminoglycosides, and macrolides (Figure 5). Interestingly, the WMS sequenced the highest percentage (41.8%) of the total ARG reads encoding resistance to tetracyclines, followed by 27.4%, 15.1%, 12%, and 1.81% of the total reads encoding resistance to macrolides, beta-lactamases, lincosamides, and aminoglycosides, respectively, while <1% of the total ARG reads encoded resistance to the rest of the antibiotic classes (Figure 6). Collectively, these data show that each dog harbored a diverse array of ARGs, potentially serving as a reservoir of diversity of antimicrobial resistance.

The host gut with its residing bacterial community is considered an important reservoir of ARGs (resistome) where gut commensals and pathogens often share the ARGs encoded by mobile genetic elements [84]. The 4560 ARG signatures (representing 48,267 ARG-specific reads) identified in this study were associated with 224 unique bacterial species spanning 103 bacterial genera (Figure 7, Appendix A). Predominantly, 82.06% of the total 4560 ARG signatures were detected in non-pathogenic bacterial genera, with 32% of the total ARG signatures associated with *Bacteroides*, followed by 12.15% of the ARG signatures detected in unknown bacterial genera (Figure 7, Appendix A). In the context of pathogenic bacterial species, 47.4% (18 out of 38 bacterial pathogens) carried ARG signatures encoding resistance to at least one class of antibiotic (Appendix A). This accounted for 5.7% (261/4560) of the ARG signatures in the pathogenic bacterial genera.

While the detected ARGs were associated with almost half of the pathogen species, not all the dogs with pathogen signatures were positive for ARGs. ARGs were detected in *Clostridium spiroforme* (7% [1/15] of positive dogs), *Clostridioides difficile* (20% [2/10] of positive dogs), *Clostridium botulinum* (17% [one of six] of positive dogs), *Enterobacter cloacae* (100% [one of one] of positive dogs), *Salmonella enterica* (100% [one of one] of positive dogs), *Streptococcus agalactiae* (25% [one of four] of positive dogs), *Streptococcus pasteurianus* (50% [one of two] of positive dogs), *Campylobacter jejuni* (25% [3/12] of positive dogs), *Clostridium perfringens* (27% [3/11] of positive dogs), *Enterococcus hirae* (56% [five of nine] of positive dogs), *Proteus mirabilis* (100% [two of two] of positive dogs), *Streptococcus gallolyticus* (20% [6/30] of positive dogs), *Streptococcus suis* (75% [three of four] of positive dogs), *Trichuris trichiura* (38% [three of eight] of positive dogs), *Staphylococcus epidermidis* (67% [two of three] of positive dogs), *Staphylococcus aureus* (40% [two of five] of positive dogs), *Enterococcus faecium* (4% [1/27] of positive dogs), and *E. coli* (63% [19/30] of positive dogs) (Appendix A).

The fecal WMS resistance profiling in this study highlights the prevalence and diversity of the antibiotic resistance genes (ARGs) carried by shelter dogs. Research shows that pet dogs often share their antimicrobial resistance (AMR) profiles with their owners, indicating potential transmission of AMR within households [85,86,87]. The resistome detected in this study was predominantly associated with non-pathogenic gut microflora with less than 20% ARGs identified in various categories of pathogenic species. The role of shelter dogs in maintaining and disseminating antimicrobial resistance remains unclear. Clinical decision-making should carefully align with current antibiotic stewardship guidelines, which may require follow-up culture and phenotypic antimicrobial susceptibility testing [88]. Nevertheless, our findings underscore the importance of ongoing monitoring and surveillance to mitigate the potential health risks for both dogs and humans.

## 4. Conclusions

This study demonstrates that shelter dogs harbor several known and potential pathogens of importance to canines, humans, and other animal species. The presence of ARGs in fecal samples suggests a potential role in the persistence and spread of antimicrobial resistance, which could contribute to treatment failures. Both zoonotic pathogens and ARGs pose a significant public health risk to shelter workers and adopters. WMS offers a valuable, comprehensive screening tool to detect a battery of pathogens and ARG carriage in dynamic animal populations such as shelter dogs which can serve as sentinels of zoonotic and other infectious disease, enabling evidence-based strategies for preventing, treating, and controlling pathogen transmission.

## Figures and Tables

**Figure 1 pathogens-14-00033-f001:**
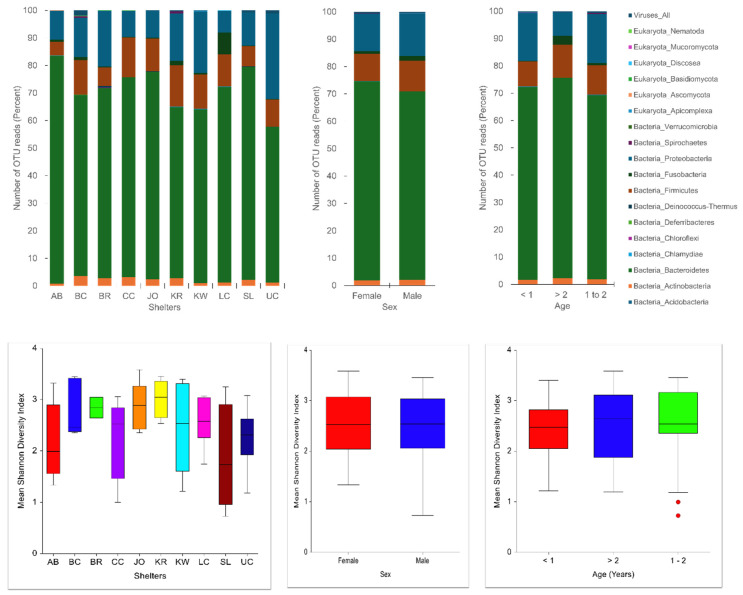
Fecal microbiome of 58 shelter dogs by shelter (AB, BC, BR, CC, JO, KR, KW, LC, SL, and UC), sex, and age (years). Bar plots (**top**) show the relative abundance of operational taxonomic units (OTU) of different bacterial phyla, eukaryotic phyla, and viruses, and the box plots (**bottom**) show alpha diversity as measured by the Shannon diversity index (SDI).

**Figure 2 pathogens-14-00033-f002:**
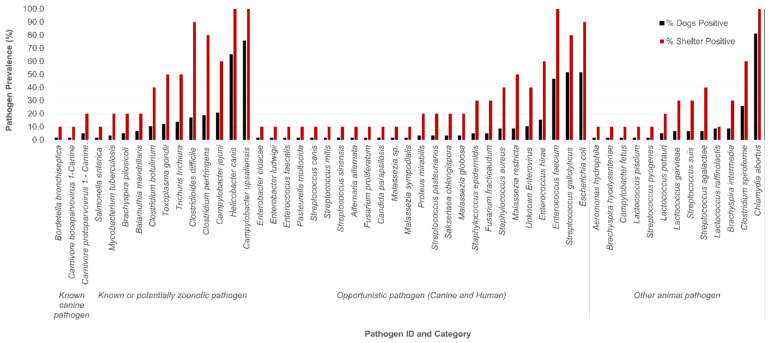
Fecal pathogen prevalence in 58 healthy dogs from 10 different shelters. Bars show the relative prevalence of bacterial, protozoal, fungal, and viral pathogen signatures. A total of 53 pathogen signatures were detected in fecal samples.

**Figure 3 pathogens-14-00033-f003:**
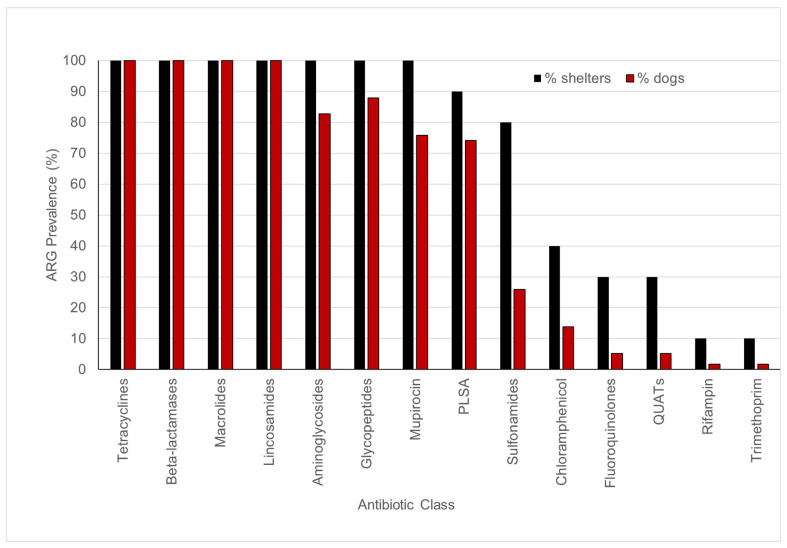
Prevalence (%) of antimicrobial resistance genes (ARGs) encoding resistance against different classes of antibiotics in fecal samples from fifty-eight dogs from 10 shelters. PLSA, pleuromutilin, lincosamide, and streptogramin A; QUATs, quaternary ammonium compounds.

**Figure 4 pathogens-14-00033-f004:**
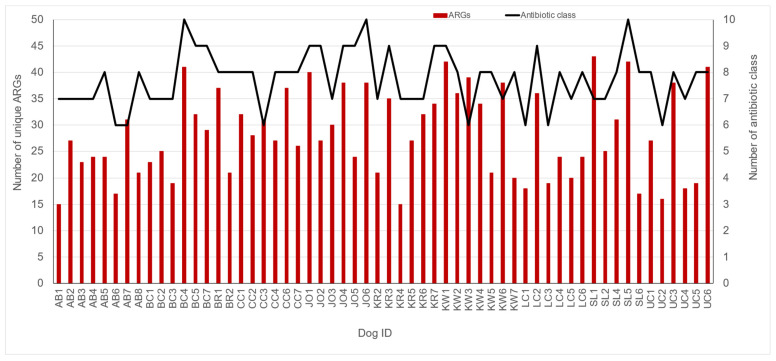
Antimicrobial resistance genes’ (ARGs) prevalence in fecal samples from fifty-eight dogs from ten shelters. Each bar represents the number of unique ARGs detected, whereas the line represents the number of antibiotic classes detected.

**Figure 5 pathogens-14-00033-f005:**
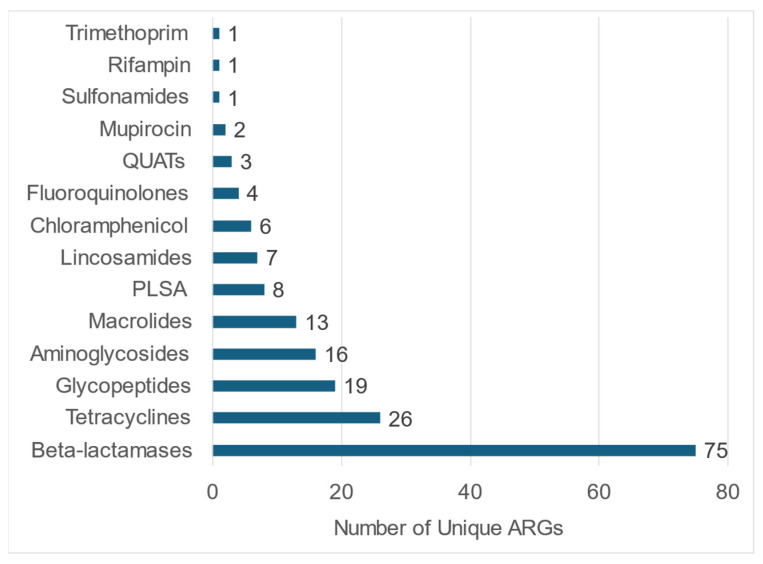
Number of unique antimicrobial resistance genes (ARGs) detected for each antibiotic class.

**Figure 6 pathogens-14-00033-f006:**
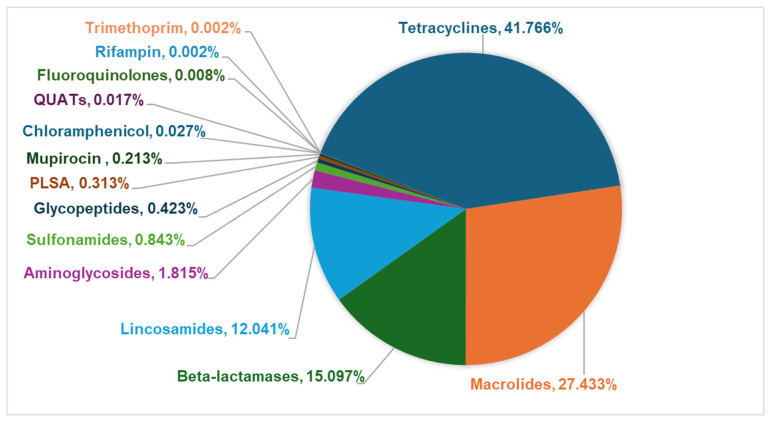
Abundance (%) of antimicrobial resistance gene (ARG) reads for each antibiotic class.

**Figure 7 pathogens-14-00033-f007:**
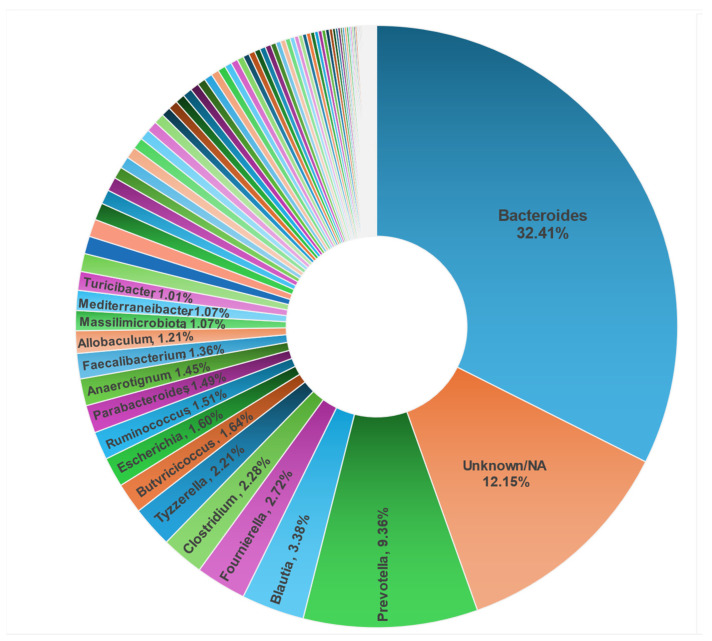
Abundance (%) of antimicrobial resistance gene (ARG) signatures associated with 103 bacterial genera in the fecal metagenome from fifty-eight dogs from ten shelters. Abundance of ARG signatures for genera ≤ 1% are shared in Appendix A.

## Data Availability

The data presented in this study are available within the article or Appendix A. The raw sequence reads associated with this study are available in the NCBI Sequence Read Archive (SRA) under BioProject ID PRJNA1197100, with BioSample numbers ranging from SAMN45772830 to SAMN45772887.

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
