# Peer review of "Pathogen Detection and Resistome Analysis in Healthy Shelter Dogs Using Whole Metagenome Sequencing"

_pathogens, 2025, doi:10.3390/pathogens14010033_

Round 1
Reviewer 1 Report
Comments and Suggestions for Authors
Title: Pathogen Detection and Resistome Analysis in Healthy Shelter Dogs Using Whole Metagenome Sequencing
Summary: This study presents and analyzes whole metagenome sequencing data from 58 canine fecal samples collected in 10 shelters in the US. The overall strengths of this study include the writing quality, the cutting-edge technology used, and the findings.
General comments:
This is a very well-written manuscript. Contributing to providing data that may improve the biosecurity and management of animals housed in shelters is needed and always a well received initiative. The screening of the dogs by whole metagenomic sequencing (WMS) showed interesting findings. Reporting the identification of potential opportunistic pathogens, including some of zoonotic importance, may contribute to building knowledge of the canine gut microbiome. This microbiome can be aMected by stressful situations, such as when animals are housed in shelters.
Some important aspects should be addressed by the authors:
1. Although the use of cutting-edge technology (WMS) is remarkable, and it would be the future of research and medical diagnostics, this tool also has several limitations. These limitations are not addressed in the manuscript. I would recommend reviewing: https://doi.org/10.1093/cid/cix881 One of my major concerns with these tools is expressed by Steve Miller and Charles Chiu: “While many of these additional organism detections correlate with the expected pathogen spectrum based on patient presentations, there are relatively few formal studies demonstrating whether these are true-positive infections and benefits to clinical outcomes”, from: https://doi.org/10.1093/clinchem/hvab173
2. Reporting antibiotic resistance genes (ARGs) from the gut can also be controversial. Other than discussing that the presence of these genes can contribute to AMR due to the possibility of transmission among gut microorganisms, the limitations of the technique/data should also be included when discussing the interpretation of these results. See: https://doi.org/10.2460/javma.23.12.0687
3. The “categories” created to classify some of the microorganisms reported can be challenging. See additional comments.
4. The authors could consider adding to the discussion some recommendations for how these findings can be used in shelters (I support the idea of expanding the idea the authors are using “screening tool”), and by veterinarians, personnel working on infection control and prevention, or researchers. What is the impact on the veterinary community? Maybe build a baseline to investigate more and make sense of these results? Some of this were added at the end of the introduction; maybe expanding these ideas.
Additional comments
Lines 87-90 Was a preservative used? The description indicates the sample was submitted to the lab in a bag only. Some publications indicate the use of a preservative may be important while the sample is transported to the lab.
Line 124 Maybe change category (i) known canine pathogens to known canine-only pathogens. This is a minor suggestion to avoid confusion with categories (iii) and (iv).
Line 185 Bordetella bronchiseptica can cause disease in other animals, especially in horses. Should this be reclassified into (iii)?
Line 208 This category is hard because a significant number of the gut microbiome members are opportunistic pathogens. Their finding in healthy dogs may not be significant at all. I celebrate the authors add some comments indicating this when describing some of the microorganisms. Maybe add a strong general comment at the beginning of the paragraph starting this section.
Lines 240-241 H. canis is occasionally reported as a “potential” cause of …. (this is still questionable)
Line 327 Opportunistic pathogens: it is hard to classify all the microorganisms included in this section as “opportunistic pathogens”. Most of them are normal members of the gut microbiota. Enterococcus spp, is expected to be there, if it is not…that may be a serious problem too. Likewise, E. coli, and other agents included in this category. We want these dogs to have these microorganisms in their gut to have a healthy microbiome. Is there a chance the authors can change this name? describe it explaining the concept of healthy microbiome? The way the authors describe these findings sends a message that these results are necessarily concerning and may cause alarm for shelters and public health environments
I would also recommend adding the limitations and potential pitfalls of WGS to the introduction.
Author Response
Authors’ Response:
We thank the reviewer for timely and thorough review, and thoughtful and constructive suggestions. Please find below our responses to each one of those comments.
General comments:
This is a very well-written manuscript. Contributing to providing data that may improve the biosecurity and management of animals housed in shelters is needed and always a well received initiative. The screening of the dogs by whole metagenomic sequencing (WMS) showed interesting findings. Reporting the identification of potential opportunistic pathogens, including some of zoonotic importance, may contribute to building knowledge of the canine gut microbiome. This microbiome can be aMected by stressful situations, such as when animals are housed in shelters.
Some important aspects should be addressed by the authors:
- Although the use of cutting-edge technology (WMS) is remarkable, and it would be the future of research and medical diagnostics, this tool also has several limitations. These limitations are not addressed in the manuscript. I would recommend reviewing: https://doi.org/10.1093/cid/cix881 One of my major concerns with these tools is expressed by Steve Miller and Charles Chiu: “While many of these additional organism detections correlate with the expected pathogen spectrum based on patient presentations, there are relatively few formal studies demonstrating whether these are true-positive infections and benefits to clinical outcomes”, from: https://doi.org/10.1093/clinchem/hvab173
Authors’ response: Thank you for your insightful comments and for bringing these important references to our attention. We have reviewed the suggested literature and incorporated it into the manuscript to provide readers with a better perspective. Specifically, we acknowledge the challenges of interpreting pathogen signatures and the need for supplementary validation methods to confirm their clinical relevance, as highlighted in shared references. In the revised manuscript, we have taken a cautious approach by emphasizing that our findings are primarily focused on pathogen detection and risk prediction, rather than clinical diagnosis.
To address reviewer’s concern, we have added the text to the Results and Discussion section: See lines, 366-373. Additionally, we have updated text in the Conclusion section: See lines, 458-460.
- Reporting antibiotic resistance genes (ARGs) from the gut can also be controversial. Other than discussing that the presence of these genes can contribute to AMR due to the possibility of transmission among gut microorganisms, the limitations of the technique/data should also be included when discussing the interpretation of these results. See: https://doi.org/10.2460/javma.23.12.0687
Authors’ response: We agree and as such we originally intended to infer in line with the reviewer's recommendation. Thus, we have avoided a claim that presence is associated with transmission. To further clarify in the revised version, we have included additional text to the Results and Discussion along with the reference. See lines, 444-450.
- The “categories” created to classify some of the microorganisms reported can be challenging. See additional comments.
Author’s response: We agree with the reviewer’s observation. In our initial classification for the detected pathogens, we explored several approaches to group them into a manageable number of categories that would be easy to follow without imposing rigid compartmentalization. While some organisms could reasonably fit into multiple categories, we carefully considered their relevance and significance to this study to assign classifications that align with our objectives and facilitate understanding. We acknowledge that there is no single ideal method to summarize such complex information comprehensively. Therefore, we have retained the current categorization for consistency and clarity.
- The authors could consider adding to the discussion some recommendations for how these findings can be used in shelters (I support the idea of expanding the idea the authors are using “screening tool”), and by veterinarians, personnel working on infection control and prevention, or researchers. What is the impact on the veterinary community? Maybe build a baseline to investigate more and make sense of these results? Some of this were added at the end of the introduction; maybe expanding these ideas.
Authors Response: Thank you. We have edited our results, discussion and conclusions to provide the clinical relevance of the findings and added alongside our responses to comment 1 and 2.
Additional comments
Lines 87-90 Was a preservative used? The description indicates the sample was submitted to the lab in a bag only. Some publications indicate the use of a preservative may be important while the sample is transported to the lab.
Authors' response: Yes, there are several approaches for sample collection and transport. For this study, samples were freshly collected and transported to the laboratory on ice within a few hours, which eliminated the need for preservatives, which in some cases have been reported to impact the microbiome of the sample.
Line 124 Maybe change category (i) known canine pathogens to known canine-only pathogens. This is a minor suggestion to avoid confusion with categories (iii) and (iv).
Authors Response: Please see our response to comment 3.
Line 185 Bordetella bronchiseptica can cause disease in other animals, especially in horses. Should this be reclassified into (iii)?
Authors' response: Thank you for this valuable suggestion. During our initial classification, we categorized Bordetella bronchiseptica under "canine-specific pathogens." However, acknowledging its broader host range, as correctly noted by the reviewer, we chose the classification "known canine pathogens" to reflect its significance as a major pathogen in dogs.
Line 208 This category is hard because a significant number of the gut microbiome members are opportunistic pathogens. Their finding in healthy dogs may not be significant at all. I celebrate the authors add some comments indicating this when describing some of the microorganisms. Maybe add a strong general comment at the beginning of the paragraph starting this section.
Authors' response: We agree and in our original manuscript, we attempted to address this organism by organism. As suggested by the reviewer, we further highlight this in section 3.2.2 (Known or potentially zoonotic pathogens). See lines, 218-220.
Lines 240-241 H. canis is occasionally reported as a “potential” cause of …. (this is still questionable)
Authors' response: We agree, and we intended to reflect this fact in our original submission. We have revised the statement accordingly for clarity. See lines 251-253.
Line 327 Opportunistic pathogens: it is hard to classify all the microorganisms included in this section as “opportunistic pathogens”. Most of them are normal members of the gut microbiota. Enterococcus spp, is expected to be there, if it is not…that may be a serious problem too. Likewise, E. coli, and other agents included in this category. We want these dogs to have these microorganisms in their gut to have a healthy microbiome. Is there a chance the authors can change this name? describe it explaining the concept of healthy microbiome? The way the authors describe these findings sends a message that these results are necessarily concerning and may cause alarm for shelters and public health environments
Authors' response: We agree. Any organism has the potential to cause infection if introduced into normally sterile areas of the body. In this sense, all organisms can be considered opportunistic pathogens. However, we focused our description on select members of the microbial community that are more commonly reported as causes of secondary infections in dogs, reflecting this context in our analysis. We have revised the statement for clarity to emphasize this distinction. The classification of "opportunistic pathogens" is most appropriate in this case, as it effectively conveys the intended meaning. Further clarification has been added in Section 3.2.3 (opportunistic pathogen). See lines, 349-356
I would also recommend adding the limitations and potential pitfalls of WGS to the introduction.
Authors Response: Please see response to comment 1 and 2.

Reviewer 2 Report
Comments and Suggestions for Authors
The manuscript addresses a critical area of research, focusing not only on the identification of pathogenic microorganisms but also on their antimicrobial resistance (AMR) gene profiles.
The literature review is comprehensive, providing the reader with a solid understanding of the topic. The materials and methods section is detailed and employs state-of-the-art methodologies for the analyses. The presentation of the results is clear and organized.
However, several points require clarification or correction:
- Regarding the sequencing raw data cutoff, why were reads shorter than 70 bp excluded? Typically, reads shorter than 200 bp are excluded as a general rule. Please provide justification for this threshold.
- Figure 1: Each figure should be interpretable on its own. Therefore, abbreviations on the horizontal axis of the bar charts in the leftmost column should be fully spelled out in the figure legend.
- In Figures 4-6, the abbreviation "ARG" should also be fully spelled out, as it was in Figure 3.
- When analyzing the ARG profile, I noticed the absence of data on the proportions of these genes located on plasmids, phages, or other mobile genetic elements. Including this information would greatly enhance the value of the manuscript.
- The discussion section is notably brief and more akin to a conclusion in its current form. It should be expanded and restructured. It appears that the authors have discussed their findings in individual results subsections, which would be better placed within the dedicated discussion section.
- The manuscript lacks the following required statements: Funding, Institutional Review Board Statement, Informed Consent Statement, and Data Availability Statement. These need to be completed at the end of the manuscript.
With these improvements, I believe this manuscript presents a timely, informative, and engaging study that will be suitable for publication.
Author Response
Authors’ Response:
We thank the reviewer for timely review and suggestions. Please find below our responses to each one of those comments.
- Regarding the sequencing raw data cutoff, why were reads shorter than 70 bp excluded? Typically, reads shorter than 200 bp are excluded as a general rule. Please provide justification for this threshold.
Authors’ Response: We used custom designed ZymoBIOMICS shotgun sequencing pipeline (Zymo Research, Irvine, CA), and ZymoBIOMICS bioinformatics analysis pipeline for this study. This pipeline utilizes Illumina NovaSeq to generate paired-end reads of 150 bp. Thus, the use of 70bp as a cut-off is reasonable. We have edited the text in materials and methods for clarity (line: 110).
- Figure 1: Each figure should be interpretable on its own. Therefore, abbreviations on the horizontal axis of the bar charts in the leftmost column should be fully spelled out in the figure legend.
Authors’ Response: Thank you. The horizontal axis of bar charts in the leftmost column are not abbreviations, but the codes (AB, BC, BR, CC, JO, KR, KW, LC, SL, and UC) that are consistently used to represent each shelter investigated in this study. We have included the code in the Figure 1 legend , and Materials and Methods (lines 85-86), as suggested by the reviewer. The detailed names of the shelters cannot be defined due to the confidentiality agreement.
- In Figures 4-6, the abbreviation "ARG" should also be fully spelled out, as it was in Figure 3.
Authors’ Response: DONE
- When analyzing the ARG profile, I noticed the absence of data on the proportions of these genes located on plasmids, phages, or other mobile genetic elements. Including this information would greatly enhance the value of the manuscript.
Authors’ Response: Thank you for highlighting this important point. The sequencing depth and platform used in our study, as well as the analytical pipeline employed, were optimized to detect and quantify ARG, but not to determine the genomic localization of genes or detection of transposon elements. Although interesting, we believe this analysis is beyond the scope of this study. Accurately determining the localization of ARGs within mobile genetic elements typically requires higher sequencing depth and specialized methodologies, such as targeted long-read sequencing (e.g., PacBio or Oxford Nanopore) or plasmidome-focused approaches, which enable the resolution of contiguity between ARGs and mobile genetic elements. Other approaches are predictive at best.
- The discussion section is notably brief and more akin to a conclusion in its current form. It should be expanded and restructured. It appears that the authors have discussed their findings in individual results subsections, which would be better placed within the dedicated discussion section.
Authors’ Response: Thank you for identifying the error in the submission file. We have corrected this error by changing the results section to “Results and conclusions” (line 134), and the discussion section to “conclusion” (line 452) as originally intended.
- The manuscript lacks the following required statements: Funding, Institutional Review Board Statement, Informed Consent Statement, and Data Availability Statement. These need to be completed at the end of the manuscript.
Authors’ Response: Information in all the sections has been added.
With these improvements, I believe this manuscript presents a timely, informative, and engaging study that will be suitable for publication.

Reviewer 3 Report
Comments and Suggestions for Authors
The authors present a thorough study of the diversity of microorganisms present in faeces of dogs from different shelters from the United States. They use whole metagenomic sequencing, a novel and powerfull technique, that allowed them to determine several pathogens at the same time. Additionally, the authors take advantage of the sequencing data and provide a resistome which is highly valuable.
Author Response
Authors’ Response:
Thank you for recognizing the value of this work. We really appreciate it.

Round 2
Reviewer 2 Report
Comments and Suggestions for Authors
Thank you very much for clarifying the rationale behind the selection of cutoff values in the Materials and Methods section.
I appreciate the corrections made to the figures, which I accept.
Although I understand the absence of plasmid, phage, and MGE investigations due to methodological limitations, it would be worthwhile to include these in future studies during sequencing efforts, as it would significantly enhance the quality of the manuscript.
I accept all other revisions as well.
The manuscript is suitable for publication.
Author Response
Authors’ Response:
Thank you for reviewing the revised version.
Comments and Suggestions for Authors
Thank you very much for clarifying the rationale behind the selection of cutoff values in the Materials and Methods section.
Thank you.
I appreciate the corrections made to the figures, which I accept.
Thank you.
Although I understand the absence of plasmid, phage, and MGE investigations due to methodological limitations, it would be worthwhile to include these in future studies during sequencing efforts, as it would significantly enhance the quality of the manuscript.
Thank you. We agree, such mechanistic investigation is a logical follow up step.
I accept all other revisions as well.
Thank you.
The manuscript is suitable for publication.
Thank you.
